# LANGUAGE MODELS FOR TEXTUAL DATA VALUATION

## ABSTRACT

In the rapidly evolving field of machine learning (ML), the quality of training data significantly impacts model performance, especially with the rise of foundation models capable of generating data. Measuring data quality may be linked to two statistical metrics: **similarity** and **diversity**, relative to a baseline dataset. We introduce *DetEmbedMetrics*, a novel deterministic embedding-based metric that enables textual data quality assessment by integrating a language model (LM) with deterministic similarity and diversity measurement functions. The core methodology constrains LM-generated embeddings to align with deterministic mathematical measurement functions, endowing the embeddings with desirable statistical properties. This approach enables the valuation of data quality by providing consistent and reliable similarity and diversity measurements, in contrast to methods directly employing neural networks for measuring data quality. Specifically, our approach involves fine-tuning a LM by inputting textual data samples with varying levels of similarity and diversity. The model learns to generate embeddings that, when applied to deterministic similarity and diversity functions, effectively capture the relationship between data sample pairs. This method allows the model to provide associated probabilities for different levels of similarity and diversity, offering clearer interpretation and decision-making compared to continuous scores. Extensive experiments on synthetic datasets demonstrate the effectiveness of DetEmbedMetrics in identifying similarity and diversity within various datasets. Notably, DetEmbedMetrics exhibits generalizability by performing robustly across different deterministic similarity and diversity functions, not relying on specific measurement techniques. This flexibility enhances its applicability as a robust framework for various measurement functions. By providing high-quality embeddings that facilitate the valuation of similarity and diversity between datasets, this research contributes to the growing field of data-centric ML, emphasizing the importance of data quality in the ML pipeline.

## 1 INTRODUCTION

Data quality remains a critical factor in machine learning (ML) success and data-driven decision-making, underpinning accurate analyses and reliable predictions across various artificial intelligence (AI) applications Gudivada et al. (2017); Jain et al. (2020); Budach et al. (2022). With the advent of language models (LMs) and large language models (LLMs), the demand for high-quality data has intensified, as it significantly influences the success of model training, performance, capabilities, and fairness Chu et al. (2024); Agiza et al. (2024); Lee et al. (2023). Consequently, data valuation, the process of measuring data quality, has gained prominence Iliou et al. (2015). This crucial step in LMs and LLMs development allows researchers and developers to rigorously examine data quality and extract high-quality subsets from raw datasets. By employing data valuation techniques, developers can better ensure the success of model training, ultimately creating more robust and reliable AI systems that perform effectively across a wide range of tasks and domains.

Evaluating the quality of data is crucial for understanding how it contributes to the performance of ML models. Kwon & Zou (2023) introduced the Data-OOB approach, which involves partitioning the data into training and unseen portions, feeding the unseen data into the trained model to see the model's performance and selecting good data for the unseen data. Alternatively, Ghorbani & Zou (2019) developed data Shapley, a method rooted in cooperative game theory, to measure each

sample's marginal contribution to the model's performance by considering how its inclusion or exclusion affects the outcome. This offers a more balanced valuation of data quality. To address its computational challenges, Chhabra et al. (2024) proposed an influence estimation technique that approximates each sample's impact on the model without requiring full retraining, making it more efficient for large-scale datasets. Although these approaches could efficiently find good data samples, they are still too computationally expensive, because it involves the model training process, and their complexity grows exponentially with the number of training data samples. Furthermore, their reliance on a test dataset for evaluating data importance may limit their practicality.

Another crucial data valuation method, without the need to train a model, involves analyzing statistical properties such as similarity and diversity Yang et al. (2024). By analyzing the statistical properties of the dataset, the model training process could be skipped and enhance the efficiency of data valuation. High similarity between training and test sets ensures consistent pattern learning and prediction. Conversely, diversity in the dataset exposes models to varied scenarios, thereby enhancing generalization and mitigating overfitting. Striking an optimal balance between these factors leads to the construction of datasets that maintain task relevance while incorporating sufficient variability. This approach results in models with improved generalization capabilities, thus enhancing their efficacy in real-world applications where data distributions may deviate from training conditions Whang & Lee (2020); Liu et al. (2016).

Amiri et al. (2023) introduces a task-agnostic approach for valuing data in marketplaces, which focuses on measuring similarity and diversity by utilizing statistical properties without needing direct access to the seller's raw data. In a related effort, Dan Friedman & Dieng (2023) introduced the Vendi score, a metric designed to assess diversity within datasets by examining the eigenvalues of a similarity matrix. This allows the Vendi score to evaluate diversity without relying on external reference datasets. Meanwhile, Charfi et al. (2020) proposed a similarity metric, InfoSpecificity, a method that combines traditional similarity metrics with information specificity, which allows it to measure the similarity not only on a sample-level but also multiple samples-level, i.e. distribution level. In addition, their experiment suggested InfoSpecificity still performs well when the data is incomplete or of low quality.

While data valuation research has yielded numerous methodologies applicable to diverse data types, the predominant training data for LMs and LLMs is textual. Consequently, textual data analysis is of significant importance in this domain. Textual data offers rich semantic content but presents unique valuation challenges due to its unstructured nature Lebart et al. (1997); Bernard & Ryan (1998). Traditional approaches, such as term frequency (TF) and n-gram are common for textual data analysis Sintia et al. (2021); Stefanovič et al. (2019). However, it's clear that while the former methods offer simplicity and effectiveness in certain cases, they come with significant drawbacks. TF simply counts how often words appear in a document, which ignores word order and relationships, leading to high-dimensional, sparse vectors that are computationally inefficient. N-grams add some structure by capturing word sequences, but they are limited by fixed-length patterns and fail to capture long-range dependencies and deeper semantic meaning. Moreover, as n-gram size increases, the feature space grows rapidly, leading to computational inefficiencies and overfitting on small datasets Almeida & Xexéo (2019); Johnson et al. (2024).

Another approach for textual data analysis, word embeddings, overcomes these issues by representing words as dense, low-dimensional vectors that encode both syntactic and semantic relationships. These embeddings, such as those from Word2Vec Church (2017) or GloVe Pennington et al. (2014), allow for words with similar meanings to be closer in vector space, capturing their contextual relationships in ways TF and n-grams cannot. By learning from large datasets, word embeddings can model more nuanced language understanding, including the dynamic meanings of words depending on their context Selva Birunda & Kanniga Devi (2021); Asudani et al. (2023).

After converting textual data into embeddings, various statistical methods can assess attributes like similarity and diversity. Zhang et al. (2019) developed BERTScore, which uses BERT Devlin (2018) embeddings to compare the similarity between texts in the token-level. Additionally, Lai et al. (2020) introduced statistical metrics for measuring diversity, density, and homogeneity, which quantify the variation, compactness, and uniformity, respectively, within datasets without label information. Although the existing methods can effectively evaluate data quality by either assessing a sample's contribution to model performance or analyzing the statistical properties of the data, they have limitations. These approaches often require access to the raw dataset or rely on word embeddings to

measure similarity and diversity, but they lack explainability, making it challenging to assess whether the embeddings are suitable for statistical evaluation.

To overcome these limitations, we focus on measuring similarity and diversity within textual datasets using embeddings generated by LMs. This approach aligns with the practical requirements of data marketplaces, where statistical properties of data, rather than raw data, are often exchanged. Consequently, this research directly contributes to data valuation in such markets and advances task-agnostic methods for textual data valuation. Moreover, we emphasize generating *high-quality embeddings* to quantify similarity and diversity—two key metrics for assessing data quality. This focus is motivated by a fundamental research question:

- *How can we provide interpretable explanations for similarity and diversity measurements between different datasets, particularly when utilizing complex LMs?*

This inquiry stems from LMs' inherent complexity as deep learning models with intricate, often difficult-understandable mechanisms. Our research aims to address this challenge by developing a framework mitigating explainability issues and yielding human-interpretable outcomes.

**Our contribution.** Our research presents a novel, deterministic embedding-based metric, *DetEmbedMetrics*, for evaluating statistical **similarity** and **diversity** in textual data. This approach combines LM-derived embeddings with deterministic functions to quantify statistical relationships between datasets. While acknowledging inherent limitations in explainability, our method aims to provide a more structured approach to textual data quality assessments.

Specifically, we strive to generate high-quality embeddings that align with the operation of deterministic mathematical functions measuring similarity and diversity by combining LM outputs with these functions. The resulting embeddings, constrained to adhere to the mathematical rules of the deterministic similarity and diversity measurement functions, exhibit desirable properties of these functions that facilitate data quality valuation tasks. As a result, these refined embeddings can be utilized to measure similarity and diversity across textual datasets using established measurement approaches. Moreover, our method addresses limitations in utilizing LLMs directly to measure relationships between textual data, particularly when dataset sizes exceed LLM input token limitations or when computational costs are prohibitive. By converting bounded textual datasets into optimized embeddings and measuring relationships using deterministic similarity and diversity functions, we circumvent these constraints, offering a more scalable and efficient approach to data quality assessment. Furthermore, the proposed methods help with identifying relevance and diverse data sources that potentially enhance model generalization.

**Notation.** Let $\mathcal{D} = \{\mathcal{D}_1, \mathcal{D}_2, \ldots, \mathcal{D}_n\}$ denote a dataset of $n$ textual samples, where each sample $\mathcal{D}_i = \{d_0(i), d_1(i), \ldots, d_N(i)\}$ consists of $N$ disjoint texts, each with a specific relationship to the text $d_0(i)$. We further collect texts with the same relationship to $d_0(i)$ and denote them with the set $\mathcal{D}_i = \{d_i(1), \ldots, d_i(n)\}$, for $i = 0, \ldots, N$. In this paper, we focus on the case where $N = 3$.

To formalize the relationships between texts in each set, we define three types of relationships relative to the collection $\mathcal{D}_0$. $\mathcal{D}_0$: A collection of base texts, each with arbitrary content. $\mathcal{D}_1$: A collection of texts, preserving the same content as in the collection $\mathcal{D}_0$ but expressed with different linguistic structures. $\mathcal{D}_2$: A collection of texts related to $\mathcal{D}_0$, differing in specific content but remaining within the same general domain or topic. $\mathcal{D}_3$: A collection of texts unrelated to $\mathcal{D}_0$, belonging to completely different domains with no direct connection to $\mathcal{D}_0$.

For the purposes of this paper, the relationships between $\mathcal{D}_0$, $\mathcal{D}_1$, $\mathcal{D}_2$, and $\mathcal{D}_3$ are evaluated using deterministic similarity and diversity metrics outlined in later sections. The similarity and diversity relationships are formalized as: similarity$(\mathcal{D}_0, \mathcal{D}_1)$ = same, similarity$(\mathcal{D}_0, \mathcal{D}_2)$ = related, similarity$(\mathcal{D}_0, \mathcal{D}_3)$ = unrelated, and diversity$(\mathcal{D}_0, \mathcal{D}_1)$ = no difference, diversity$(\mathcal{D}_0, \mathcal{D}_2)$ = diverse, diversity '$(\mathcal{D}_0, \mathcal{D}_3)$ = totally different.

The evaluation of similarity and diversity between four different texts can be framed as a three-class classification task, with two sets of fixed labels: $[1, 0, 0]$, $[0, 1, 0]$, and $[0, 0, 1]$, respectively. Similarity and diversity are measured separately but in parallel, using two deterministic functions: one for assessing similarity and the other for evaluating diversity. It is important to note that while the labels for similarity and diversity are numerically identical, they have different meanings. For similarity, the first element in the label refers to the pair being the same, the second to being related,

and the third to being unrelated. In contrast, for diversity, the first element indicates no difference, the second indicates diversity, and the third refers to being completely different.

**Paper Arrangement.** **Section 2** formulates the problem and introduces our proposed approach. **Section 3** describes DetEmbedMetrics' methodology, including deterministc similarity and diversity measurement functions, model architecture, and training process. **Section 4** presents the experimental setup and results, comparing DetEmbedMetrics with benchmarks and assessing its robustness and generalizability. **Section 5** provides a comprehensive summary of the paper's key findings and contributions and explores potential avenues for future research and extensions of this work.

## 2    PROBLEM STATEMENT AND THE PROPOSED APPROACH

Accurately assessing data quality is crucial for effective ML training, impacting model performance. Evaluating statistical properties like similarity and diversity (Amiri et al., 2023) is one approach, but measuring these between textual datasets poses challenges. Existing methods often miss semantic nuances, lack embedding explainability, or require full data access, raising privacy concerns Bernard & Ryan (1998); Ghorbani & Zou (2019); Zhang et al. (2019). To address these issues, we introduce DetEmbedMetrics, which generates high-quality embeddings to efficiently quantify similarity and diversity. This approach preserves privacy and offers a more interpretable method for assessing data quality, mitigating embedding inexplicability.

Specifically, we combine LM embeddings with deterministic similarity and diversity functions to compare textual datasets. This integration appends these functions to the LM embedding layer, deriving similarity and diveristy scores from functions rather than directly from embeddings. Joint optimization of LM and the deterministic functions during training generates embeddings capturing dataset similarity and diversity. This aligns embeddings with the deterministic functions' characteristics, enabling effective relationship capture. Our method yields discrete classes with probabilities for similarity and diversity, contrasting with continuous, potentially ambiguous scores.

Next, we will discuss the merits of combining LM with deterministic similarity and diversity functions, and the merits of converting continuous scores to discrete classes with associated probabilities.

**Combination of LM and Deterministic Similarity and Diversity Measurement Functions.** ML methods, particularly deep learning models like LMs, are often criticized for their lack of interpretability. Directly employing LMs to measure the similarity and diversity between two datasets might exacerbate this issue due to the complexity of LMs. While combining LM embeddings and deterministic similarity and diversity measurement functions does not inherently resolve the fundamental inexplicability of the LM embeddings, it offers potential minor gains in interpretability, which is due to its more structured framework for analysis compared to using raw LM outputs. This approach strikes a balance between leveraging the representational power of neural networks and applying more transparent mathematical techniques for the final data quality assessment.

Applying the LM embeddings to the deterministic functions for measuring similarity and diversity ensures consistent comparisons between embeddings, even if the embeddings themselves remain opaque. Furthermore, it allows for the analysis of relative relationships between datasets, potentially offering insights into dataset structures.

**Conversion from Continuous Scores to Discrete Classes with Associated Probabilities.** Having discrete classes with associated probabilities addresses several drawbacks with continuous scoring:

- Interpretability challenges: Human interpretation of small numerical differences in similarity or diversity scores is often difficult. For instance, with cosine similarity scores ranging from [0, 1], distinguishing the practical significance between scores of 0.8 and 0.75 is challenging. These subtle differences may not reflect meaningful distinctions in real-world applications, complicating decision-making based on such scores.
- Ambiguity in score definition: Assigning scores to items with intermediate similarity or diversity introduces ambiguity. Consider texts $\mathcal{D}_0$ (original), $\mathcal{D}_1$ (a rephrased version of $\mathcal{D}_0$), $\mathcal{D}_2$ (related to $\mathcal{D}_0$ but with different content), and $\mathcal{D}_3$ (unrelated to $\mathcal{D}_0$). In terms of similarity, when using cosine similarity as metrics, scoring $(\mathcal{D}_0, \mathcal{D}_1)$ as 1 indicates high similarity, while scoring $(\mathcal{D}_0, \mathcal{D}_3)$ as 0 reflects complete dissimilarity. However, determining a score for $(\mathcal{D}_0, \mathcal{D}_2)$, which shares a related topic but differs in specific content,

is less straightforward. This ambiguity in intermediate scoring complicates analysis and extends to diversity measurements as well. The challenge lies in consistently quantifying relationships between datasets that range from homogeneous to highly varied, potentially leading to inconsistencies in analysis.

Figure 1 provides an illustration of the proposed approach with these classes for similarity and diversity, corresponding to three levels of each characteristic.

Here we demonstrate DetEmbedMetrics' effectiveness in capturing similarity and diversity between various dataset pairs with a motivating exaple. Consider a LM pre-trained on $\mathcal{D}_0$ European Medieval art-related textual data (paintings and sketches). We evaluate four potential fine-tuning datasets: $\mathcal{D}_1$ (European Medieval marble statues), $\mathcal{D}_2$ (Chinese Medieval paintings), $\mathcal{D}_3$ (highly-relevant European Medieval paintings), and $\mathcal{D}_4$ (Modern sports data). We measure similarity and diversity between $\mathcal{D}_0$ and each dataset, classifying as [same, related, unrelated] and [no difference, diverse, totally different] respectively. DetEmbedMetrics yields:

- $(\mathcal{D}_0, \mathcal{D}_1)$: similarity = [0.1, 0.6, 0.3], diversity = [0.1, 0.8, 0.1].
- $(\mathcal{D}_0, \mathcal{D}_2)$: similarity = [0.2, 0.7, 0.1], diversity = [0.2, 0.7, 0.1].
- $(\mathcal{D}_0, \mathcal{D}_3)$: similarity = [0.8, 0.2, 0.0], diversity = [0.7, 0.3, 0.0].
- $(\mathcal{D}_0, \mathcal{D}_4)$: similarity = [0.0, 0.1, 0.9], diversity = [0.0, 0.2, 0.8].

Results suggest $\mathcal{D}_1$ and $\mathcal{D}_2$ introduce beneficial diversity while maintaining relevance. $\mathcal{D}_1$ is less similar to $\mathcal{D}_0$ than $\mathcal{D}_2$, reflecting the shift from painting to sculpture versus geographical change. $\mathcal{D}_1$ introduces slightly more diversity. Both remain within the art domain, offering valuable fine-tuning information. $\mathcal{D}_3$'s high similarity and low diversity indicate limited new information potential. $\mathcal{D}_4$, unrelated to $\mathcal{D}_0$, diverges too far for useful fine-tuning. These results confirm intuitive dataset pair connections, showing the approach's effectiveness in capturing similarity and diversity traits.

## 3 METHODOLOGY

In this section, we present a detailed description of DetEmbedMetrics, including the deterministic similarity and diversity measurement functions, the model architecture, and the training process.

### 3.1 THE DETERMINISTIC SIMILARITY AND DIVERSITY MEASUREMENT FUNCTIONS

DetEmbedMetrics employs deterministic functions to measure similarity and diversity using embeddings of datasets pairs. In the following, we describe these functions.

**Similarity Measurement Function.** We use Manhattan distance to measure similarity between embeddings. For two vectors $\boldsymbol{x}, \boldsymbol{y} \in \mathbb{R}^n$ with $i$-th entries $x_i$ and $y_i$, respectively, we have:

$$d(\boldsymbol{x}, \boldsymbol{y}) = \sum_{i=1}^{n} |x_i - y_i|. \tag{1}$$

This metric sums the absolute differences across dimensions, effectively comparing vector representations in high-dimensional spaces.

**Diversity Measurement Function.** To evaluate the diversity of the embeddings, we employ Vendi score (Dan Friedman & Dieng, 2023), which offers a flexible and general approach to quantifying diversity in ML contexts, addressing the limitations of domain-specific metrics or those requiring reference datasets. Given a vector $\boldsymbol{x} = [x_1, \ldots, x_n]$, the Vendi score is defined as:

$$\mathrm{VS}_k(\boldsymbol{x}) = \exp\Big( -\sum_{i=1}^{n} \lambda_i \log \lambda_i \Big), \tag{2}$$

where $\lambda_1, \ldots, \lambda_n$ are the eigenvalues of $\boldsymbol{K}/n$, and $\boldsymbol{K}$ is an $n \times n$ kernel matrix with entries $K_{i,j} = k(x_i, x_j)$ for a user-defined similarity function $k$.

### 3.2 THE ARCHITECTURE OF DETEMBEDMETRICS

DetEmbedMetrics integrates LMs with the above deterministic functions to evaluate similarity and diversity between dataset embeddings. Conventional methods often employ average pooling to summarize information across the entire sequence length of LM outputs, with shape $[B, L, H]$, where

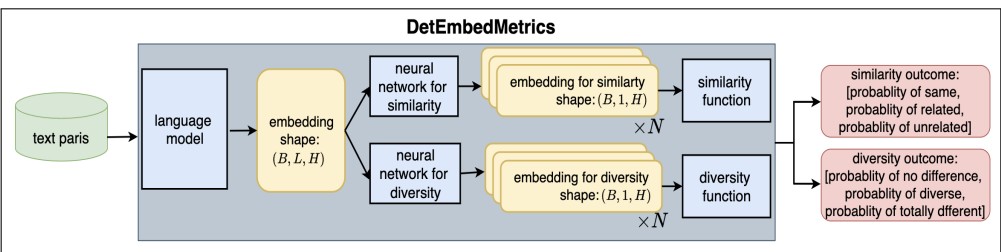

Figure 1: DetEmbedMetrics architecture: This system processes text pairs and generates probability distributions for similarity (same, related, or unrelated) and diversity (no difference, diverse, or totally different) classes. Specifically, the architecture generates $N$ embeddings for the similarity and diversity metrics, respectively, with each embedding designed to capture a specific aspect of the textual relationship. This multi-embedding approach facilitates a comprehensive representation of textual relationships, simultaneously capturing various dimensions of similarity and diversity.

$B$ denotes the batch size, $L$ represents the sequence length, and $H$ is the hidden state dimension, and then utilize the average-pooled embedding, with shape $[B, 1, H]$, for downstream tasks. Our experiments show this approach inadequate for model training. An intuitive explanation is that our goal is to generate discrete classes rather than continuous scores. While deterministic functions measure statistical relationships between LM embeddings effectively, they yield continuous scores. Converting these to discrete classes, as noted earlier, is challenging, impeding model training.

To overcome this limitation, we propose a novel framework generating $N$ embeddings from the LM output, each representing an aggregation of the whole sequence. The number $N$ corresponds to the classification task's class count, with each embedding learning a specific relationship. Notably, DetEmbedMetrics generates two $N$-embedding sets—one for similarity, one for diversity—enabling parallel yet separate measurements.

To demonstrate DetEmbedMetrics, consider a three-class problem ($N = 3$) evaluating relationships between texts $\mathcal{D}_0$, $\mathcal{D}_1$, $\mathcal{D}_2$, and $\mathcal{D}_3$, focusing on similarity (analogous for diversity). We quantify similarity between pairs $(\mathcal{D}_0, \mathcal{D}_1)$, $(\mathcal{D}_0, \mathcal{D}_2)$, and $(\mathcal{D}_0, \mathcal{D}_3)$, with corresponding labels $[1, 0, 0]$, $[0, 1, 0]$, and $[0, 0, 1]$, respectively. These one-hot encoding labels represent "the same", "related," and "unrelated" categories, where the first embedding measures the degree of identity between the pair, the second embedding quantifies the extent of relatedness, and the third embedding assesses the degree of unrelatedness.

By dedicating separate embeddings to each relationship category, we enable the model to learn and represent complex textual relationships more effectively. It is important to note that, with DetEmbedMetrics, the NN is to generate suitable embeddings, which align with the operation of the deterministic measurement functions, while the operation of these functions remains fixed and uses consistent mathematical logic to measure the similarity and diversity between datasets.

Figure 1 illustrates the complete model architecture for our proposed approach. The flowchart demonstrates that the embeddings from the LM undergo further processing for the downstream task. Specifically, these embeddings serve as input to two small NNs, each outputing $N$ distinct embeddings, each of which encapsulates information about the entire sequence and is designed to capture different aspects of similarity and diversity relationships.

In addition, to ensure probabilistic outputs, we normalize results by dividing each class probability by the sum of all probabilities, yielding valid distributions across relationship categories. Similarity and diversity functions then measure relationships using these specialized embeddings. For example, with $N = 3$ for similarity, both datasets have 3 embeddings representing [same, related, unrelated]. The similarity function uses the first embedding pair to measure dataset identity, the second for relatedness, and the third for unrelatedness. Diversity measurement follows the same process with its own specialized embeddings.

Next, we describe the training process, which aims to train the model to generate embeddings that can capture the similarity and diversity among pairs of data with the desired granularity after being applied to the respective deterministic functions.

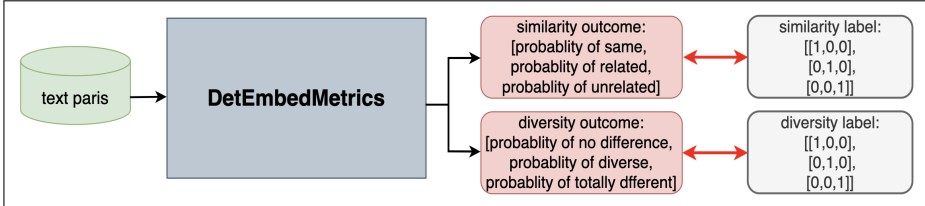

Figure 2: This figure illustrates the concept behind using three classes. The model's objective is to generate predictions that approximate a diagonal matrix as closely as possible.

### 3.3 TRAINING PROCESS

The training aim is to generate embeddings with statistical properties aligned with deterministic similarity and diversity functions, performed separately but in parallel as shown in Figure 1. Textual data is converted into two embedding types—from similarity and diversity NNs, respectively. These embeddings measure similarity and diversity against other textual data embeddings. For example, similarity NN embedding pairs (e.g., from $\mathcal{D}_0$ and $\mathcal{D}_1$) fed into the deterministic similarity function output discrete classes with probabilities: [same, related, unrelated]. This process repeats for ($\mathcal{D}_0$, $\mathcal{D}_2$) and ($\mathcal{D}_0$, $\mathcal{D}_3$), producing relationship vectors. Similarly, diversity embeddings yield vectors with [no difference, diverse, totally different] probabilities, representing relationships between $\mathcal{D}_0$ and $\mathcal{D}_1$, $\mathcal{D}_2$, and $\mathcal{D}_3$.

We use textual datasets with relationships between pairs ($\mathcal{D}_0$, $\mathcal{D}_1$), ($\mathcal{D}_0$, $\mathcal{D}_2$), and ($\mathcal{D}_0$, $\mathcal{D}_3$) labeled $[1, 0, 0]$, $[0, 1, 0]$, and $[0, 0, 1]$ respectively. Labels for similarity and diversity are numerically identical but semantically different. During training, we feed $\mathcal{D}_0$, $\mathcal{D}_1$, $\mathcal{D}_2$, and $\mathcal{D}_3$ simultaneously, obtaining predicted classes (model output) and fixed labels for each pair. We concatenate the three predicted classes and their labels to form prediction and label matrices. The fixed labels form a diagonal matrix. The predicted probabilities (prediction matrix) and corresponding fixed labels for similarity are shown below:

$$\textbf{Prediction} = \begin{bmatrix} p_{\text{same}}(\mathcal{D}_0, \mathcal{D}_1) & p_{\text{related}}(\mathcal{D}_0, \mathcal{D}_1) & p_{\text{unrelated}}(\mathcal{D}_0, \mathcal{D}_1) \\ p_{\text{same}}(\mathcal{D}_0, \mathcal{D}_2) & p_{\text{related}}(\mathcal{D}_0, \mathcal{D}_2) & p_{\text{unrelated}}(\mathcal{D}_0, \mathcal{D}_2) \\ p_{\text{same}}(\mathcal{D}_0, \mathcal{D}_3) & p_{\text{related}}(\mathcal{D}_0, \mathcal{D}_3) & p_{\text{unrelated}}(\mathcal{D}_0, \mathcal{D}_3) \end{bmatrix}, \quad \textbf{Label} = \begin{bmatrix} 1 & 0 & 0 \\ 0 & 1 & 0 \\ 0 & 0 & 1 \end{bmatrix}.$$

Here, $p_{\text{same}}$, $p_{\text{related}}$, and $p_{\text{unrelated}}$ are predicted probabilities for each class (same, related, unrelated) for respective text pairs. The fixed label matrix shows true relationships: $\mathcal{D}_0$ and $\mathcal{D}_1$ are same, $\mathcal{D}_0$ and $\mathcal{D}_2$ related, and $\mathcal{D}_0$ and $\mathcal{D}_3$ unrelated. Diversity measurement follows the same procedure.

The diagonal structure allows us to treat the label matrix as an image-like representation, where the diagonal pixels are 1, and the rest are 0. To minimize the difference between the prediction matrix and the label matrix, we employ the Sinkhorn distance (Cuturi, 2013) as the loss function. The Sinkhorn distance, a refined version of the Wasserstein distance (Kantorovich, 2006), evaluates the difference across all elements of the matrices, unlike cross-entropy, which only considers the element corresponding to the correct label. This enables the Sinkhorn distance to effectively measure the discrepancy between two probability distributions. Our experimental results also indicate that using cross-entropy as the loss function fails to train the model effectively. Figure 2 illustrates this concept for a three-class problem ($N = 3$), where the goal is for the model's predictions to approximate the diagonal matrix as closely as possible.

## 4 EXPERIMENTS

We conducted three experiments to evaluate DetEmbedMetrics' efficacy, comparative performance, and generalization capabilities.

**Experiment 1 (Comparison with Benchmark LMs).** This experiment compares our method, DetEmbedMetrics, with widely-used LMs. We utilize the alternative LMs without training or fine-tuning, directly employing their embeddings to compute similarity and diversity scores. We employ Manhattan distance for similarity and Vendi score for diversity assessment, given in equation 1 and equation 2, respectively, comparing this baseline approach with DetEmbedMetrics.

**Experiment 2 (Various Deterministic Similarity and Diversity Measurement Functions).** We assess DetEmbedMetrics' robustness using alternative deterministic functions for similarity and diversity measurements. This experiment tests whether the model maintains performance across different measures, indicating methodological effectiveness beyond a single set of measurements. Consistent performance would evidence the approach's robustness and generalizability, suggesting broader applicability in textual data quality assessment.

**Experiment 3 (Cross-Domain Generalizability Assessment).** This experiment tests DetEmbed-Metrics' performance on an unseen dataset from a different domain (e.g., training on art, evaluating on sports). We assess the model's cross-domain generalization, both with and without domain-specific training data. Strong performance without additional training demonstrates robust generalization. If suboptimal, we introduce minimal new domain data. Significant improvement with limited new data would indicate the model's general ability to distinguish texts, regardless of domain. This design evaluates the model's capacity to discern content, structure, and semantic differences generalizably, and its ability to leverage limited domain data for enhanced performance.

### 4.1 SYNTHETIC DATA GENERATION

While our approach is conceptually straightforward, acquiring an appropriate dataset for training is challenging. We used LLaMA-3 8B (AI@Meta, 2024) and GPT-4o (OpenAI, 2024) to generate a synthetic dataset. We instructed these LLMs to generate four paragraphs ($\mathcal{D}_0$, $\mathcal{D}_1$, $\mathcal{D}_2$, and $\mathcal{D}_3$) with specific relationships with easy examples: $\mathcal{D}_0$ (Arbitrary content): "I love dogs so much.", $\mathcal{D}_1$ (Paraphrase of $\mathcal{D}_0$): "I am a dog person.", $\mathcal{D}_2$ (Content related to $\mathcal{D}_0$): "I pet 3 cats, and they are the treasure in my life.", $\mathcal{D}_3$ (Content entirely different from $\mathcal{D}_0$): "I majored in archaeology and am now a well-known archaeologist.".

Here, $\mathcal{D}_0$ and $\mathcal{D}_1$ convey the same sentiment with slight structural variations. $\mathcal{D}_0$ and $\mathcal{D}_2$ share a pet theme, while $\mathcal{D}_3$ diverges significantly.

Using LLMs for synthetic data generation offers cost-reduction benefits in data collection and pre-processing. To mitigate potential homogeneity in grammatical structures from a single LLM, we employed both LLaMA-3 8B and GPT-4o. In our third experiment, we trained DetEmbedMetrics on LLaMA-3 8B's art-related data and evaluated it on sports-related data from both LLMs, testing cross-domain and language pattern generalization. The generation prompt and sample data are in the Appendix.

### 4.2 EXPERIMENT 1: COMPARING WITH BENCHMARK LMS

We generated 9,900 samples using LLaMA-3 8B, with 7,920 for training and 1,980 for testing. We modified "all-mpnet-base-v2" (Song et al., 2020)[1] by incorporating a small NN as shown in Figure 1. This modified model serves as DetEmbedMetrics' LM. For benchmarking, we selected two popular Hugging Face LMs:

- Unmodified "all-mpnet-base-v2" model.
- "BAAI/bge-large-en-v1.5" model. (Xiao et al., 2023)

For benchmarks, we used zero-shot learning, utilizing pre-trained embeddings without fine-tuning. These embeddings were inputs for the similarity (Manhattan distance) and diversity (Vendi score with Manhattan distance kernel) metrics. Table 1 shows performance on 1,980 unseen test samples, revealing significant performance disparity between benchmarks and our approach.

Table 1: The results show that DetEmbedMetrics has apparent improvement than the benchmark LMs in terms of accuracy, precision, recall, and F1 scores (both macro and micro).

| Model | Relation | Accuray | Precision macro | Precision micro | Recall macro | Recall micro | F1 macro | F1 micro |
|---|---|---|---|---|---|---|---|---|
| DetEmbedMetrics | similarity | 1.00 | 1.00 | 1.00 | 1.00 | 1.00 | 1.00 | 1.00 |
| | diversity | 0.97 | 0.97 | 0.97 | 0.97 | 0.97 | 0.97 | 0.97 |
| all-mpnet-base-v2 | similarity | 0.31 | 0.11 | 0.31 | 0.31 | 0.31 | 0.16 | 0.31 |
| | diversity | 0.67 | 0.50 | 0.67 | 0.67 | 0.67 | 0.56 | 0.67 |
| BAAI/bge-large-en-v1.5 | similarity | 0.33 | 0.11 | 0.33 | 0.33 | 0.33 | 0.17 | 0.33 |
| | diversity | 0.33 | 0.11 | 0.33 | 0.33 | 0.33 | 0.17 | 0.33 |

---

[1]We use "sentence-transformers/all-mpnet-base-v2", fine-tuned for sentence similarity tasks.

### 4.3 EXPERIMENT 2: VARIOUS DETERMINISTIC SIMILARITY AND DIVERSITY MEASUREMENT FUNCTIONS

To assess our approach's consistency and generalizability, we expanded experiments beyond initial deterministic functions. Originally, we used Manhattan distance for similarity and Vendi score (with Manhattan distance kernel) for diversity. Our goal was to determine if performance generalizes across various deterministic measures. We note that "VS with Manhattan" refers to the use of Vendi score as the diversity measurement function, with Manhattan distance serving as the kernel function.

Table 2 shows extended evaluation results. Findings demonstrate robust performance across different deterministic similarity and diversity functions, indicating our approach's generalizability. However, Euclidean distance as similarity function and Vendi score kernel caused numerical instabilities. This suggests that while robust, not all measurement functions suit our methodology equally.

Table 2: Performance of DetEmbedMetrics across various similarity and diversity functions. The table shows accuracy, precision, recall, and F1 scores (both macro and micro) for similarity and diversity relationships using different deterministic functions.

| Similarity Function | Diversity Function | Relationship | Accuracy | Precision macro | Precision micro | Recall macro | Recall micro | F1 macro | F1 micro |
|---|---|---|---|---|---|---|---|---|---|
| Manhattan Distance | VS with Manhattan | similarity | 1.00 | 1.00 | 1.00 | 1.00 | 1.00 | 1.00 | 1.00 |
| | | diversity | 0.97 | 0.97 | 0.97 | 0.97 | 0.97 | 0.97 | 0.97 |
| Cosine Similarity | VS with Cosine Similarity | similarity | 0.99 | 0.99 | 0.99 | 0.99 | 0.99 | 0.99 | 0.99 |
| | | diversity | 0.97 | 0.97 | 0.97 | 0.97 | 0.97 | 0.97 | 0.97 |
| CKA (Klabunde et al., 2024) | VS with CKA | similarity | 0.99 | 0.99 | 0.99 | 0.99 | 0.99 | 0.99 | 0.99 |
| | | diversity | 1.00 | 1.00 | 1.00 | 1.00 | 1.00 | 1.00 | 1.00 |
| AngShape (Klabunde et al., 2024) | VS with AngShape | similarity | 1.00 | 1.00 | 1.00 | 1.00 | 1.00 | 1.00 | 1.00 |
| | | diversity | 0.98 | 0.98 | 0.98 | 0.98 | 0.98 | 0.98 | 0.98 |

### 4.4 EXPERIMENT 3: CROSS-DOMAIN GENERALIZABILITY ASSESSMENT

To evaluate cross-domain generalizability, we conduct a two-step experiment. Step one involves incrementally adding data from one domain to determine the data quantity needed for stable, effective performance. We train the model on an unmodified "all-mpnet-base-v2" augmented with a small NN, as shown in Figure 1. For each data increment, we train from scratch and evaluate on a fixed validation dataset.

Step two evaluates the trained model from step one on an unseen dataset from another domain. We incrementally add new domain training data to the pre-trained model, determining the quantity needed for good performance on the new validation set. We also monitor performance on the original validation set to ensure consistency. Our aim is to achieve good performance in the new domain without compromising effectiveness in the original domain, maintaining cross-domain generalizability. Next, we provide the experimental details of each step.

**Step one.** we incrementally train DetEmbedMetrics on art-related data from LLaMA-3 8B (with training set sizes ranging from 32 to 2080 samples), assessing performance on a fixed 1000-sample validation set.

**Step two.** using the model trained on 2080 art samples, we gradually add sports data (0 to 648 samples), evaluating on both art and sports validation sets (1000 samples each). We use sports datasets from LLaMA-3 8B and GPT-4o to account for language pattern differences. We also test baseline cross-domain generalizability on sports data without sports-specific training to observe if the model could perform well on a different content domain without additional training.

Figure 3 presents the results. The first row shows enhanced performance with increased art-related training data, with incremental accuracy improvements for similarity and diversity beyond about 600 samples. The second and third rows demonstrate improved sports domain performance while maintaining art-related task consistency when incorporating sports data (from LLaMA-3 8B and GPT-4o, respectively). Notably, adding relatively few sports samples (about 40 from LLaMA-3 8B and 168 from GPT-4o, while the model was originally trained on samples from LLaMA-3 8B) suffices for good generalization to the new domain. Comprehensive results are in the Appendix. Results are averaged over 10 iterations to mitigate NN randomness. This experiment illustrates the model's ability to generalize to a new domain without compromising performance on the original domain, and the impact of using data from different LLMs on cross-domain generalizability.

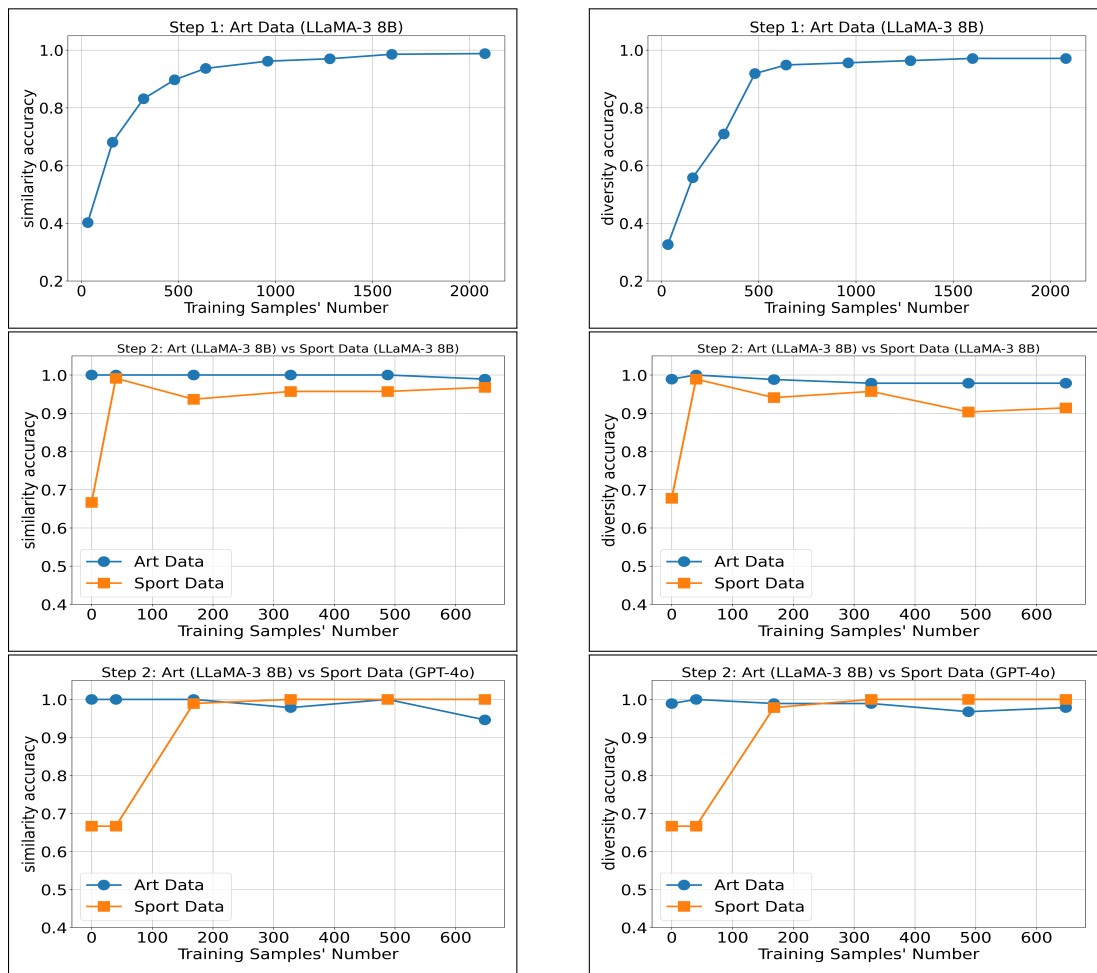

Figure 3: Performance improvement with increasing number of training samples. Top row: art domain. Middle row: sports domain (LLaMA-3 8B). Bottom row: sports domain (GPT-4o). Left column: similarity accuracy. Right column: diversity accuracy.

## 5 CONCLUSIONS

DetEmbedMetrics offers a novel textual data quality valuation approach, generating high-quality embeddings aligned with deterministic similarity and diversity functions. This two-step process—creating optimized embeddings capturing nuanced textual relationships, then applying deterministic functions—bridges LM' representational power and mathematical consistency. Experiments show DetEmbedMetrics' superior performance, robustness across various deterministic functions, and cross-domain generalizability. By providing a systematic method for embedding generation and subsequent similarity and diversity assessment, DetEmbedMetrics significantly contributes to data-centric ML, offering a powerful tool for textual dataset valuation and improvement. While not fully resolving interpretability challenges, it represents progress towards more transparent textual data quality assessment. Furthermore, depending on chosen deterministic functions, this approach may help identify embedding dimensions' contributions to similarity or diversity measures, potentially enhancing explainability. For instance, employing a deterministic similarity function that assesses similarity dimension by dimension could endow the embeddings with special properties in each dimension once the model is well-trained, thereby enhancing their explainability. While DetEmbedMetrics advances embedding interpretability, further research on LM explainability remains crucial for a deeper understanding of internal mechanics, including embedding generation processes.

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

# A  APPENDIX

## A.1  SYNTHETIC DATA GENERATION PROMPT

Here is the prompt used to generate synthetic data:

Create four distinct paragraphs, A, B, C, and D, adhering to the following conditions:

Paragraph A: Write an informative paragraph on any topic of your choice.

Paragraph B: Rephrase Paragraph A entirely, maintaining the exact same content but using a different structure and wording. The information conveyed should be identical to Paragraph A.

Paragraph C: Write a paragraph that is related to Paragraph A and within the same general domain, but focuses on different specific content. For example, if Paragraph A is about baseball, Paragraph C could be about tennis - both are sports, but they are different sub-domains. Ensure that the relationship between Paragraphs A and C is clear and logical.

Paragraph D: Create a paragraph on a topic from a completely different domain than Paragraph A. For instance, if Paragraph A is about technology, Paragraph D could be about medicine.

Note:

When generating Paragraph C, take extra care to review Paragraph A again to ensure they are appropriately related within the same general domain.

Example structure:

Domain 1 is related with tech (i.e. Quantum computing and AI), and Domain 2 is about medicine.

Paragraph A [Topic X in Domain 1]: In the fast-paced world of technology, the development of quantum computers stands as a monumental achievement. These advanced systems leverage the principles of quantum mechanics to process information at speeds unattainable by classical computers. The core of quantum computing revolves around the quantum bit, or qubit, which can exist in multiple states simultaneously, thus offering exponential growth in processing power. This capability enables quantum computers to solve complex problems, such as cryptographic challenges and optimization tasks, which are currently beyond the reach of traditional computing technologies.

Paragraph B [Rephrased version of Topic X in Domain 1]: Quantum computing represents a significant breakthrough in technology, utilizing the principles of quantum mechanics to enhance processing speeds beyond what classical computers can achieve. These systems are built around qubits, which unlike traditional bits, can hold multiple states concurrently, significantly expanding computational capacity. This unique feature allows for tackling highly complex tasks, including cryptography and optimization problems, which conventional computers find challenging.

Paragraph C [Related Topic Y in Domain 1]: Artificial intelligence (AI) systems are increasingly being integrated into various industries to optimize processes and enhance decision-making. AI algorithms are designed to analyze data and learn from patterns, enabling them to make predictions or decisions without human intervention. This technology is particularly influential in areas like finance, where it can predict market trends and automate trading activities.

Paragraph D [Unrelated Topic Z in Domain 2]: In the realm of medicine, the discovery of CRISPR gene-editing technology marks a revolutionary advancement. This innovative technique utilizes a specialized DNA sequence found in bacteria, which can be engineered to target and modify genes in other organisms. By precisely altering the genetic structure, CRISPR has the potential to treat a variety of genetic disorders by correcting mutations at their source. The application of this technology extends from simple genetic modifications to complex therapeutic interventions, tackling diseases that were once deemed incurable.

## A.2  SAMPLE FROM SYNTHETIC DATASET

Here is one example from the generated synthetic dataset, including paragraphs $\mathcal{D}_0$, $\mathcal{D}_1$, $\mathcal{D}_2$ and $\mathcal{D}_3$.

- Paragraph $\mathcal{D}_0$: During a basketball game, teamwork plays an essential role in achieving victory. Each player on the court has specific responsibilities and must communicate effectively with their teammates. This collaboration ensures that both offensive and defensive plays are executed smoothly. Additionally, understanding each member's strengths and weaknesses allows the team to strategize more effectively, maximizing their chances of winning the game.

- Paragraph $\mathcal{D}_1$: Teamwork is crucial for success in a basketball game. Every player has distinct roles and needs to communicate properly with fellow team members. This synergy helps in the efficient execution of offensive and defensive strategies. Moreover, knowing

each player's strengths and weaknesses enables better strategic planning, increasing the team's likelihood of winning.

- Paragraph $\mathcal{D}_2$: In soccer, the goalkeeper has one of the most challenging tasks on the field. Their main job is to prevent the opposing team from scoring by blocking shots on goal. A good goalkeeper needs excellent reflexes, strong decision-making skills, and the ability to stay calm under pressure. They also need to coordinate closely with the defenders to ensure that the back line remains solid and organized, making it difficult for opponents to penetrate.

- Paragraph $\mathcal{D}_3$: In the field of medical surgery, teamwork is vital for a successful operation. Each member of the surgical team has specific roles that must be clearly understood and executed with precision. Effective communication among the team members is crucial for the smooth execution of the operation procedures. Additionally, knowing each team member's specialized skills and limitations helps in crafting a precise surgical plan, thereby increasing the chances of a successful outcome for the patient.

## A.3    COMPLETE RESULTS OF EXPERIMENT 3: CROSS-DOMAIN GENERALIZABILITY ASSESSMENT

This section presents the comprehensive results of experiment 3, where we gradually added samples to observe performance changes. Note that due to identical macro and micro results for Precision, Recall, and F1, we only display macro values.

Figure 4 shows the complete results of step one, where we trained an unmodified "all-mpnet-base-v2" model augmented with a small NN (as illustrated in Figure 1). We incrementally added art-related data to observe performance improvements on the art-related validation dataset.

For step two, Figure 5 presents the results of gradually adding sport-related data generated by LLaMA-3 8B. Similarly, Figure 6 shows the results for sport-related data generated by GPT-4o. These graphs demonstrate the model's cross-domain generalization capabilities and the impact of different data sources on performance.

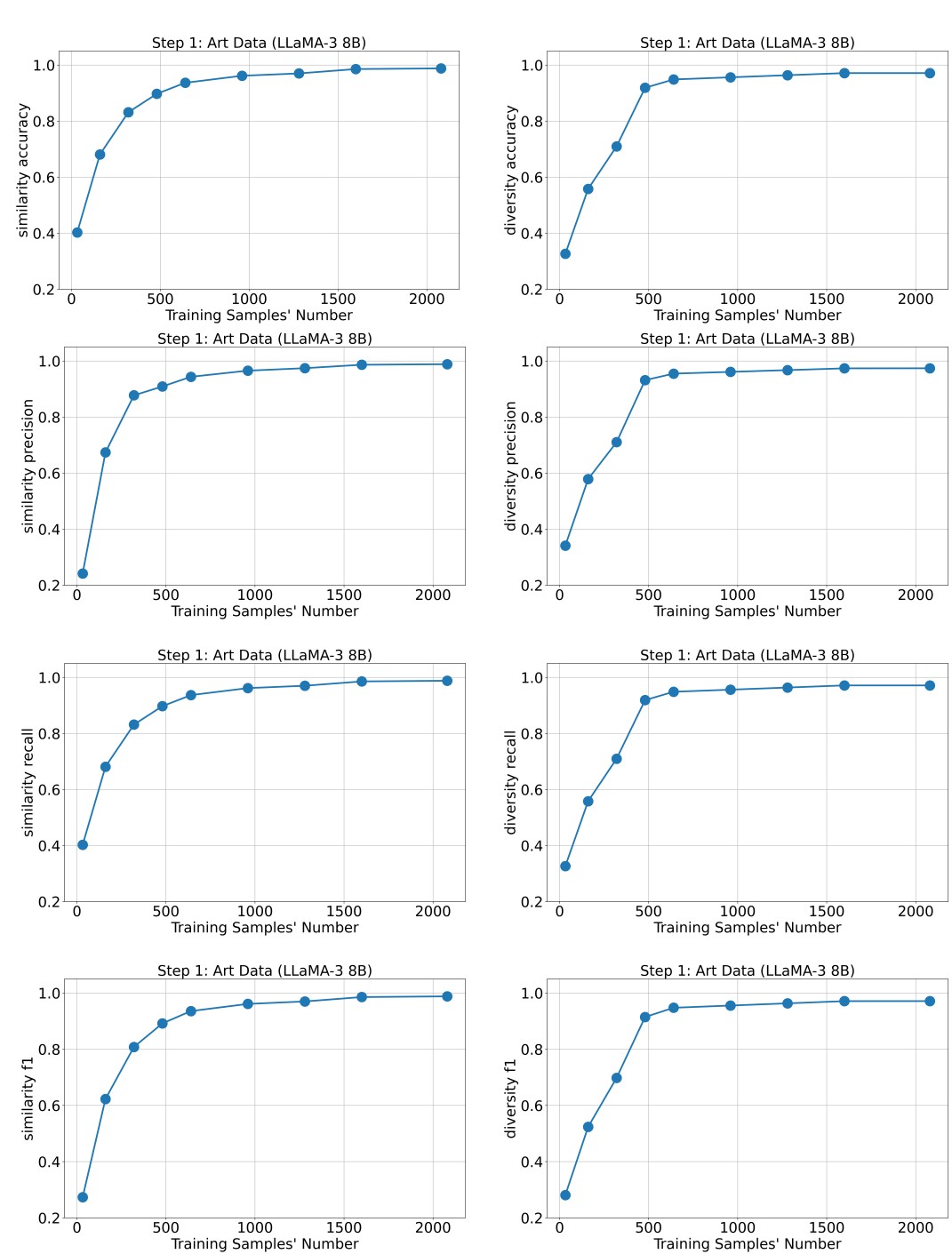

Figure 4: Complete results of step one (art data generated by LLaMA-3 8B) from experiment 3

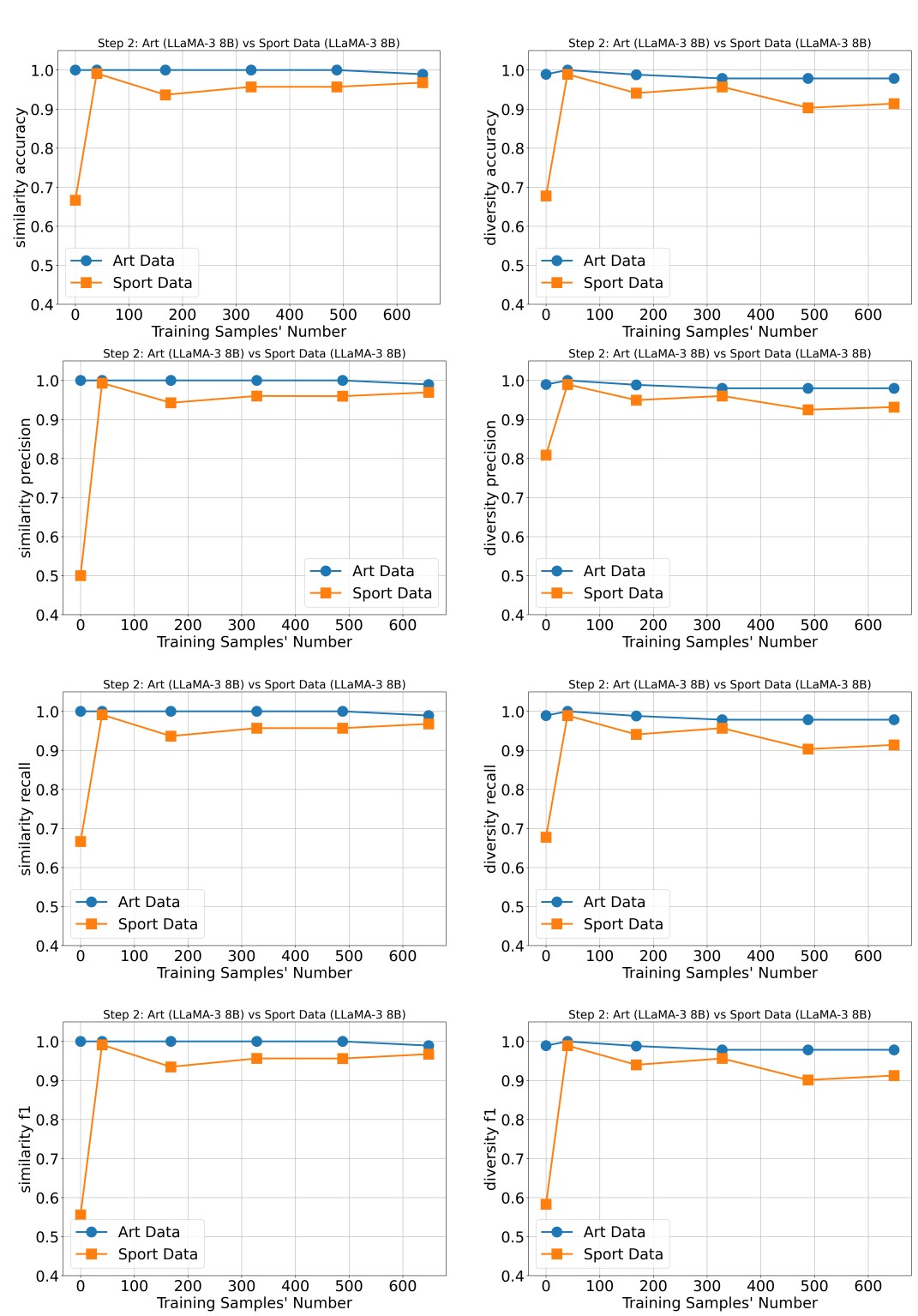

Figure 5: Complete results of step two (sport data generated by LLaMA-3 8B) from experiment 3

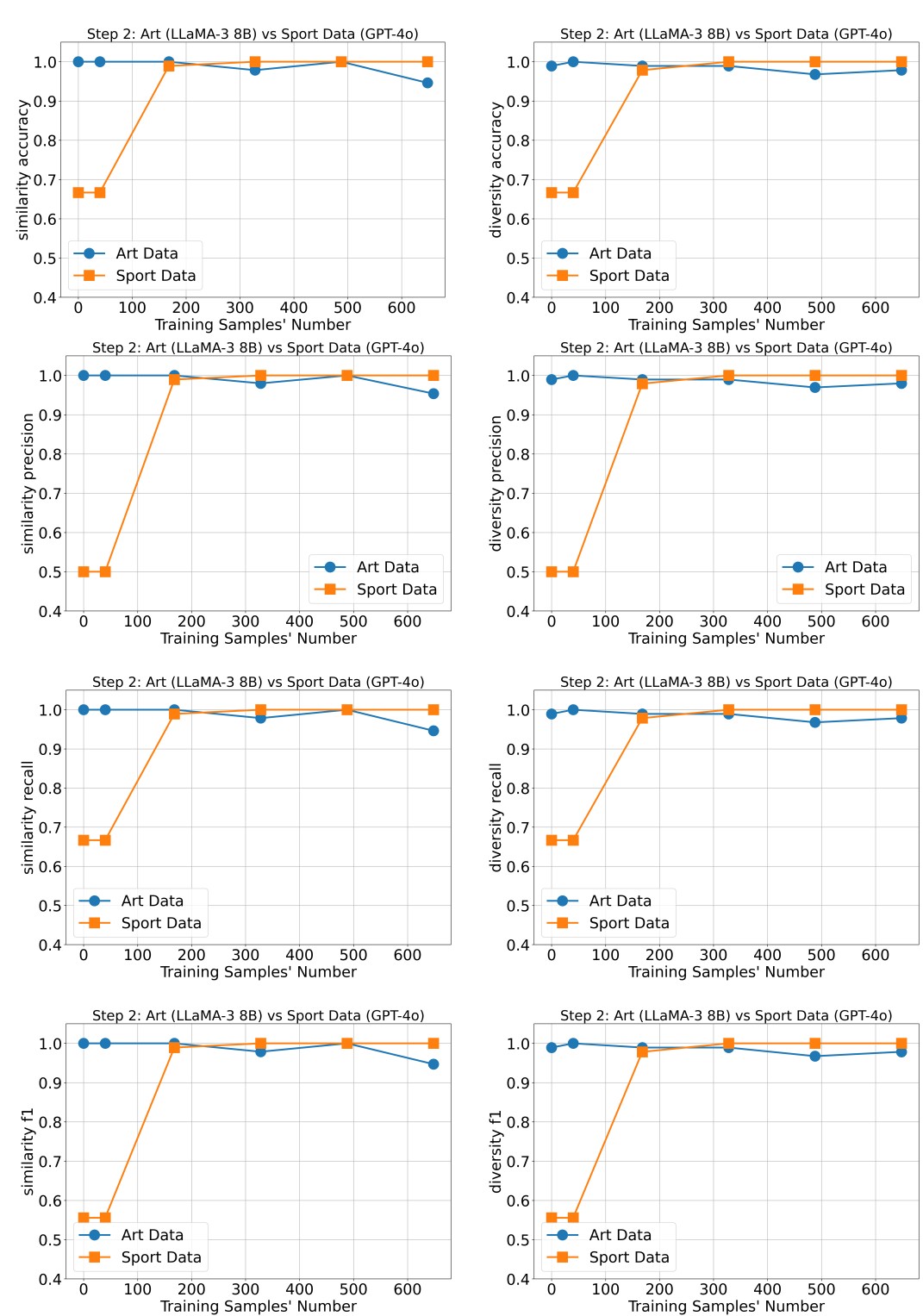

Figure 6: Complete results of step two (sport data generated by GPT-4o) from experiment 3

