# OpenReview forum: "Language Models for Textual Data Valuation"
_ICLR.cc/2025/Conference — ICLR 2025 Conference Withdrawn Submission_

### Official Review · Reviewer_UEwW · 2024-10-23

**Soundness:** 2
**Presentation:** 2
**Contribution:** 1
**Rating:** 3
**Confidence:** 3

**Summary:**

This paper introduces a method for measuring the similarity and diversity across subsets of language model training datasets. The paper frames the problem as a parallel multiclass prediction problem, and interprets the class prediction scores as probabilities. The paper provides simulated results that support the claim that their method improves upon baseline methods of similarity and diversity measurement for language model training data.

**Strengths:**

The paper presents a clear method for estimating the similarity and diversity of training datasets. The method in the paper is much more computationally tractable than existing alternatives.

**Weaknesses:**

There are several issues with the paper, primarily regarding a lack of novelty and a lack of experimental evidence.

From a novelty perspective, the authors method effectively boils down to training a classifier to determine pairwise classes in a batch of data. While there is some novelty in the exact choices used for this training process (including distance function, loss function, etc...), these choices are not examined experimentally.

Additionally, the authors opt to simply interpret the class scores of their classifier as probabilities.
The validity of this interpretation would be improved dramatically if a calibration study were present.

Experimentally, the paper's claim of improvement over baseline is supported by a study against a single baseline on a single dataset that the authors generated synthetically.
The claim would be much stronger if the authors compared against multiple baseline models on several datasets.
Ideally, comparisons would be made with existing datasets, as well as datasets synthetically generated by the authors.

Further, the authors use Manhattan distance for similarity and diversity computations of the baseline they compare against, bge-large-en-v1.5. Since bge-large-en-v1.5 canonically compares data using dot products, and this choice may significantly alter the results of the baseline. The authors provide a robustness study under different distances for their proposed method, but not for the baseline method.

**Questions:**

The notation in lines 140-144 was very confusing to me, particularly due to using i as both an argument and a subscript in your definitions.

---

### Official Review · Reviewer_LXKh · 2024-11-03

**Soundness:** 2
**Presentation:** 2
**Contribution:** 1
**Rating:** 1
**Confidence:** 5

**Summary:**

The paper proposes DetEmbedMetrics as a new metric that uses deterministic embedding-based methods to assess textual data quality by measuring similarity and diversity. The paper claims that this approach fine-tunes a language model to generate embeddings that align with deterministic functions, providing reliable data quality measurements and demonstrating effectiveness across various datasets.

**Strengths:**

1. High data quality is an important problem.
2. Authors experiment with multiple similarity and diversity metrics.

**Weaknesses:**

1. Valuation or evaluation? If it is valuation, why don't you compute the dollar value of the datasets?
2. The authors experiment  with synthetic datasets only. No results have been shown with real datasets. Why not use datasets from GLUE benchmark where there are 3 tasks (e.g., QQP) to compute similarity between sentence pairs? Similarly, why not show results on some real world diversity datasets?
3. Authors relate data valuation with quality of datasets with similarity and diversity with some reference dataset. Are there no other notions of defining dataset quality? Further, they link notions of similarity and diversity with generating embeddings. I believe that these 4 things (1) data valuation (2) similarity and diversity with some reference dataset (3) dataset quality (4) generating embeddings, are extremely loosely coupled and authors' arguments in the introduction try to stitch a story between these 4 which is very unclear. The authors show no results on evaluation of dataset quality or data valuation as such. Based on original motivation, it is expected that with better data, authors should have trained better language models providing state of the art results on many NLP tasks.
4. A poster child example in the introduction will help clarify the proposed problem better.
5. It would have been nice if you could compare various public data collections and show why some models perform better (because they have been trained with similar and/or diverse datasets) and why others with low quality datasets perform worse.
6. Fig 1: Details of the "neural network" are not provided.
7. The proposed methodology has no technical depth as such.
8. Generating synthetic dataset from LLMs is a great idea to train models, but not to evaluate models. What is the accuracy of samples which were used for testing?
9. Table 1: Essentially all-mpnet-base-v2 pretrained model is compared with finetuned model. It is widely known that finetuning improves the model -- so what is new in the results? No hyper-param details are provided (not even in the appendix).
10. Why just 3 classes for similarity? Why not compute similarity like that done in Semantic Textual Similarity Benchmark (STS-B) in GLUE?

Typos and presentation issues:
1. citep and citet are not used properly. E.g. on line 44, it should be (Chu et al. ,2024); (Agiza et al. ,2024); (Lee et al., 2023)" and not "Chu et al. (2024); Agiza et al. (2024); Lee et al. (2023)"
2. Line 142 and 144 define D_i in 2 different ways. Why are there 2 definitions? I am confused.
3. D_1, D_2 are samples. D_0 is a collection? Notations seem to make no sense to me.
4. Line 141 calls D_1 as a sample. Line 146 calls D_1 as a document collection.
5. Line 184: diveristy
6. Line 289: "show this approach inadequate"

**Questions:**

Please see weaknesses.

---

### Official Review · Reviewer_fvCw · 2024-11-04

**Soundness:** 1
**Presentation:** 2
**Contribution:** 1
**Rating:** 1
**Confidence:** 4

**Summary:**

- The authors propose a new measure to score the similarity and diversity across datasets used to train language models. A key feature is that both methods output probability vectors on $\Delta^3$, which the authors describe as “deterministic” and claim enhances interpretability.
- Experiments confirm that fine-tuning text embeddings with synthetic training data improves score prediction performance on synthetic test data compared to cases without fine-tuning.

**Strengths:**

- This paper addresses an important issue. Language models are now deeply integrated not only within natural language processing and machine learning but across society as a whole. Accurately measuring the quality of datasets used to train these models is not only technically challenging but also carries significant social impact.
- The introduction provides a thorough overview of previous research, which should serve as a helpful starting point for readers to understand the state of the field.

**Weaknesses:**

- The motivation for the “deterministic” approach is not clearly presented, nor are any empirical benefits demonstrated. While the paper argues that the deterministic nature should enhance interpretability, the shift from a score on $\mathbb{R}$ to one on $\Delta^3$ actually adds complexity. Additionally, the experimental results lack a human evaluation to assess whether interpretability has indeed improved, and there are no experiments to verify whether this method effectively aids data selection for training language models.
- The writing is generally vague, making the content difficult to interpret. For example, Section 3.2 and Figure 1 describe the proposed method, yet even the formal type of the function is not specified. As a result, reproducibility is notably reduced.

**Questions:**

N/A

---

### Official Review · Reviewer_YbjQ · 2024-11-06

**Soundness:** 2
**Presentation:** 1
**Contribution:** 2
**Rating:** 3
**Confidence:** 3

**Summary:**

This paper introduces a deterministic metric named DetEmbedMetrics for evaluating the quality of textual data using similarity and diversity measurements. The method leverages an LM to generate embeddings that are constrained by deterministic mathematical functions for similarity and diversity analysis. The authors highlight the significance of data quality in machine learning, particularly in the context of LLMs, and emphasize that their approach provides more reliable and interpretable data quality assessment compared to existing methods that rely on neural networks. By fine-tuning the LM with textual data samples of varying similarity and diversity, the method enables robust evaluation and produces probabilities associated with similarity and diversity levels. The paper also demonstrates the flexibility and generalizability of DetEmbedMetrics through extensive experiments on synthetic datasets.

**Strengths:**

1. The problem studied in this paper is interesting and important.

2. The paper presents a deterministic metric for similarity and diversity analysis.

**Weaknesses:**

1. The article is difficult to follow. It would be helpful if the author could clarify and reorganize the introduction and methodology sections for better coherence.

2. The experiments presented are not robust, and there are too few baseline comparisons. A more comprehensive set of benchmarks would strengthen the analysis.

3. The paper should include more discussions and comparisons with current literature [1,2] on data quality control in LLMs.



[1] Ge Y, Liu Y, Hu C, et al. Clustering and ranking: Diversity-preserved instruction selection through expert-aligned quality estimation[J]. arXiv preprint arXiv:2402.18191, 2024.

[2] Du Q, Zong C, Zhang J. Mods: Model-oriented data selection for instruction tuning[J]. arXiv preprint arXiv:2311.15653, 2023.

**Questions:**

n/a

---

### Note · Authors · 2024-11-20

**Comment:**

Thank you for your detailed review. We have decided to withdraw our submission to incorporate your valuable feedback and make the work more complete. We will submit an improved version that addresses your comments.

**Withdrawal Confirmation:**

I have read and agree with the venue's withdrawal policy on behalf of myself and my co-authors.